# A Novel Spectral Index for Tracking Preload Change from a Wireless, Wearable Doppler Ultrasound

**DOI:** 10.3390/diagnostics13091590

**Published:** 2023-04-29

**Authors:** Jon-Emile S. Kenny, Zhen Yang, Geoffrey Clarke, Mai Elfarnawany, Chelsea E. Munding, Andrew M. Eibl, Joseph K. Eibl, Jenna L. Taylor, Chul-Ho Kim, Bruce D. Johnson

**Affiliations:** 1Health Sciences North Research Institute, Sudbury, ON P3E 2H3, Canada; 2Flosonics Medical, Toronto, ON P3C 1R7, Canada; 3Northern Ontario School of Medicine, Sudbury, ON P3E 2C6, Canada; 4Human Integrative and Environmental Physiology Laboratory, Department of Cardiovascular Diseases, Mayo Clinic, Rochester, MN 55905, USA

**Keywords:** Doppler ultrasound, carotid artery, internal jugular vein, hemorrhage, central venous pressure

## Abstract

A wireless, wearable Doppler ultrasound offers a new paradigm for linking physiology to resuscitation medicine. To this end, the image analysis of simultaneously-acquired venous and arterial Doppler spectrograms attained by wearable ultrasound represents a new source of hemodynamic data. Previous investigators have reported a direct relationship between the central venous pressure (CVP) and the ratio of the internal jugular-to-common carotid artery diameters. Because Doppler power is directly related to the number of red cell scatterers within a vessel, we hypothesized that (1) the ratio of internal jugular-to-carotid artery Doppler power (V/A_POWER_) would be a surrogate for the ratio of the vascular areas of these two vessels and (2) the V/A_POWER_ would track the anticipated CVP change during simulated hemorrhage and resuscitation. To illustrate this proof-of-principle, we compared the change in V/A_POWER_ obtained via a wireless, wearable Doppler ultrasound to B-mode ultrasound images during a head-down tilt. Additionally, we elucidated the change in the V/A_POWER_ during simulated hemorrhage and transfusion via lower body negative pressure (LBNP) and release. With these *Interesting Images*, we show that the Doppler V/A_POWER_ ratio qualitatively tracks anticipated changes in CVP (e.g., cardiac preload) which is promising for both diagnosis and management of hemodynamic unrest.

**Figure 1 diagnostics-13-01590-f001:**
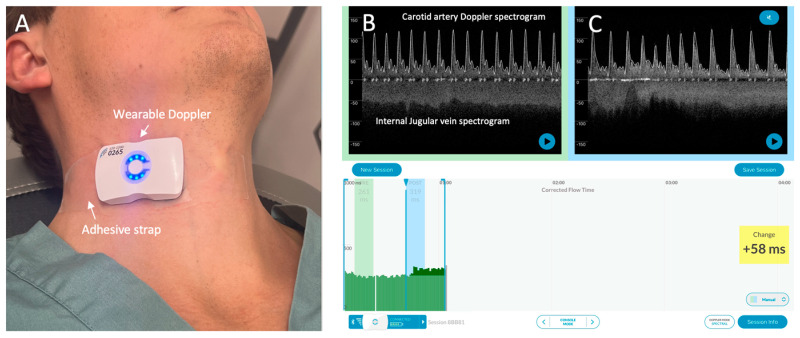
The wireless, wearable Doppler system [1,2]. (**A**) The wearable Doppler transducer adhered on a healthy volunteer. (**B**) The graphical user interface (GUI) of the Doppler system. The carotid and internal jugular spectrograms are shown in baseline section of a hemodynamic assessment. (**C**) shows the spectrograms from the same subject during a preload augmentation.

**Figure 2 diagnostics-13-01590-f002:**
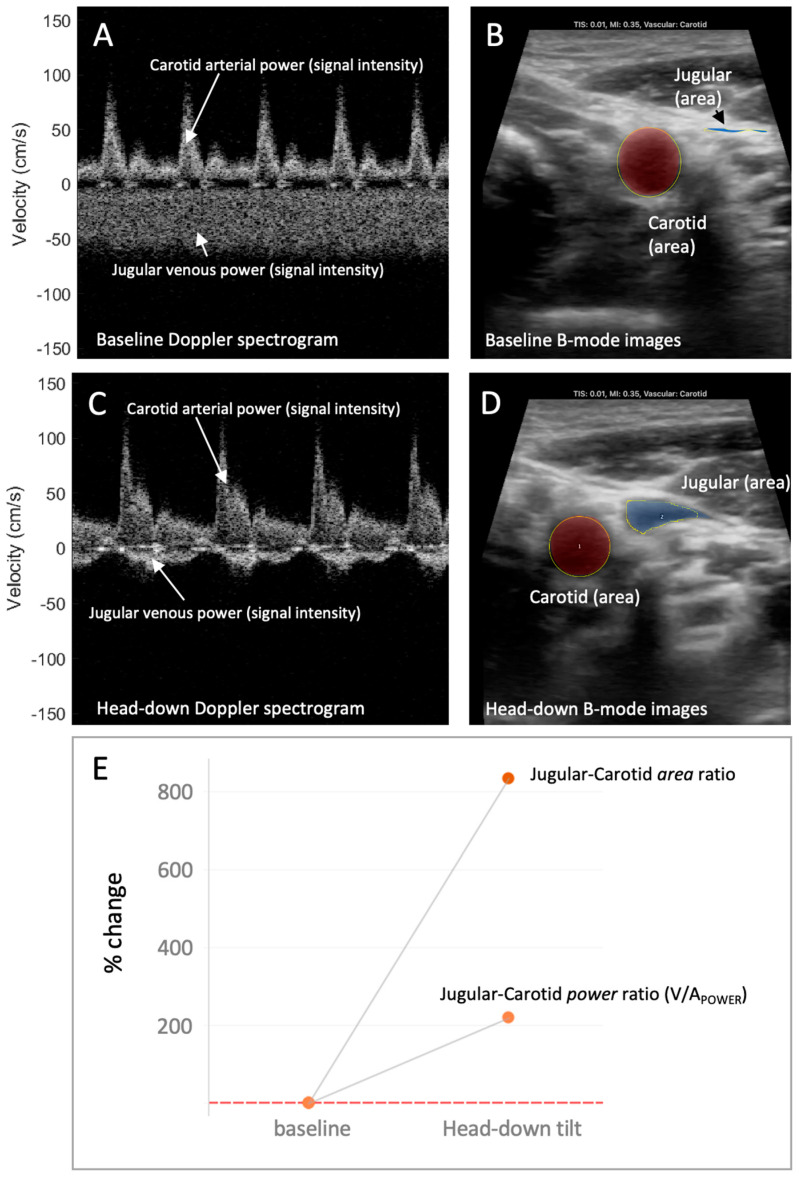
The venous–arterial power (V/A_POWER_) index and relation to venous–arterial area during increased preload. (**A**) The jugular venous and carotid arterial Doppler spectrograms during low preload condition (i.e., baseline, fully upright). The Doppler power relates to the “brightness” or “intensity” of the signal. Here, the jugular *power* (intensity) compared to the carotid power (intensity) is relatively low. (**B**) The jugular venous and carotid arterial vascular *area* during low preload condition (i.e., baseline, fully upright). Here, the jugular relative to carotid area is also low. (**C**) The jugular venous and carotid arterial Doppler spectrograms during high preload condition (i.e., head-down). The jugular *power* (intensity) relative to the carotid power (intensity) has increased. (**D**) The jugular venous and carotid arterial vascular area during high preload condition (i.e., head-down). Here, the jugular relative to carotid area has increased. (**E**) The change in the jugular–carotid (i.e., venous–arterial, V/A) Doppler power ratio and jugular–carotid (i.e., venous–arterial) area ratio from low preload (i.e., upright, baseline) to high preload (i.e., head-down) conditions. The jugular vein-to-carotid artery area ratio is a surrogate for central venous pressure [3,4].

**Figure 3 diagnostics-13-01590-f003:**
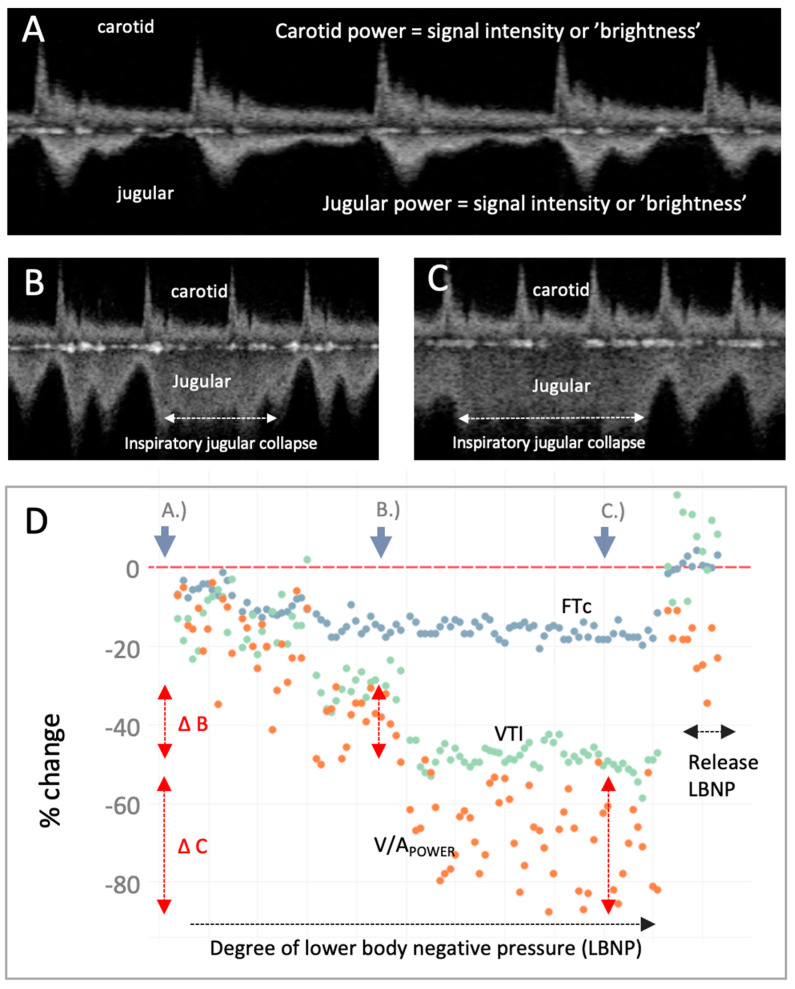
The venous–arterial power (V/A_POWER_) index during moderate-to-severe central hypovolemia and resuscitation. (**A**) Carotid and jugular spectrograms during resting, supine baseline. (**B**) Carotid and jugular spectrograms midway through the lower body negative pressure (LBNP) protocol which induces mild-to-moderate central hypovolemia. The jugular spectrogram has a reduced power relative to the carotid (i.e., decreased V/A_POWER_) consistent with decreasing jugular vein-to-carotid artery area. (**C**) Carotid and jugular spectrograms during late LBNP which induces moderate-to-severe central hypovolemia; the V/A_POWER_ has observably fallen further. Partial jugular collapse with inspiration is marked as the higher, less-distinct velocity pattern. (**D**) Plot of V/A_POWER_ (orange dots) during progressively severe LBNP and release. The portions of the LBNP protocol from which the displayed Doppler spectrograms were obtained are indicated on the graphic (resting baseline is far left, increasing central hypovolemia progresses rightwards followed by LBNP release which induces rapid central blood volume expansion). V/A_POWER_ falls as the degree of LBNP intensifies, intimating falling jugular-to-carotid area ratio (i.e., diminishing CVP), as anticipated [5]. While total V/A_POWER_ is falling, there is increasing V/A_POWER_ variability from baseline (i.e., point A.) to mild-to-moderate central hypovolemia at point B. (i.e., vertical red arrows, ∆B) to moderate-to-severe central hypovolemia at point C. (i.e., ∆C); this illustrates inspiratory collapse and expiratory distension of the jugular vein. Corrected flow time (FTc, blue dots) and velocity time integral (VTI, green dots) of the carotid artery are measures from the carotid artery Doppler spectrogram and are stroke volume (SV) surrogates [6,7,8,9]. With LBNP release (i.e., auto-transfusion), V/A_POWER_ increases consistent with rising CVP; so, too, do the measured SV surrogates. To further our conclusions, especially given known inter-observer variability when assessing vascular dimensions [10], comparing 3-dimensional vessel area [11] to V/A_POWER_ and validating this ratio against computational models [12] will improve confidence in its physiological fidelity.

## Data Availability

Data available upon reasonable request.

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
