# Peer review of "A Novel Spectral Index for Tracking Preload Change from a Wireless, Wearable Doppler Ultrasound"

_diagnostics, 2023, doi:10.3390/diagnostics13091590_

Round 1
Reviewer 1 Report
The topic is very interesting. However, some limitations of the Doppler technology need to be discussed in details to justify the scientific soundness of this paper.
1. Regarding the area ratio, it depends on the angle of imaging and is highly dependent on the operators' experience. Therefore, the area ratio is often derived from the measurement of 3D artery geometry reconstructed from radiological images (Refer: 10.3389/fcvm.2021.597568).
2. In figure 3 B and C the envelop of Jugular waveform especially during systole is vague. Please explain how did you get accurate estimation of the velocity.
3. I suggest to validate the results by comparing with the results of existing computational studies (e.g., 10.3389/fphys.2023.1085871, 10.1016/j.jbiomech.2006.07.008)
Author Response
We thank the reviewer for his or her time with our letter. We believe that we have addressed his or her criticisms below:
The topic is very interesting. However, some limitations of the Doppler technology need to be discussed in details to justify the scientific soundness of this paper.
- Regarding the area ratio, it depends on the angle of imaging and is highly dependent on the operators' experience. Therefore, the area ratio is often derived from the measurement of 3D artery geometry reconstructed from radiological images (Refer: 10.3389/fcvm.2021.597568)
We agree with the reviewer that image acquisition can be challenging and is most certainly dependent upon operator experience. We have added in a line indicating the further validation against 3D modelling would improve our future work on this topic. We have added the relevant reference as well. thank you!
2. In figure 3 B and C the envelop of Jugular waveform especially during systole is vague. Please explain how did you get accurate estimation of the velocity.
Thank you for making this clear, these indistinct areas occur when the subject breathes in, lowers jugular venous pressure and causes transient collapse of the vein, so the velocity typically gets a bit higher and indistinct. We have added this in the figure to make it more clear. Because we are focused on Doppler power, the absolute velocity is less relevant.
3. I suggest to validate the results by comparing with the results of existing computational studies
Thank you for this criticism, we have added this concern and relevant reference for the reader at the end of figure 3.
Reviewer 2 Report
The Authors presented an "Interesting Images" paper, focused on the definition of a novel spectral index from a wireless and wearable Doppler ultrasound able to track preload change.
The brief abstract description is informative enough and the images are quite interesting with clear legends.
I enjoy reading the paper.
Author Response
we thank the reviewer for his or her time with our letter.
Round 2
Reviewer 1 Report
Thanks for the update. My earlier comments have been largely addressed. However, the limitation of ultrasound method needs further clarification. Especially, the inter-observer variablity can lead to inaccuracy, which is often considered in ultrasound-based measurement (Refer: 10.1016/j.atherosclerosis.2011.05.006) as well as hemodynamic evaluation (Refer: 10.1016/j.jns.2017.08.3239). Therefore, the conclusion need to be validated on larger datasets by different observers/operators.
Author Response
we thank the reviewer for this clarification. we have amended our conclusion to make clear the importance of inter-observer variability and added the pertinent reference.